# Effect of Oyster Mushroom Addition on Improving the Sensory Properties, Nutritional Value and Increasing the Antioxidant Potential of Carp Meat Burgers

**DOI:** 10.3390/molecules28196975

**Published:** 2023-10-08

**Authors:** Grzegorz Tokarczyk, Katarzyna Felisiak, Iwona Adamska, Sylwia Przybylska, Agnieszka Hrebień-Filisińska, Patrycja Biernacka, Grzegorz Bienkiewicz, Małgorzata Tabaszewska

**Affiliations:** 1Department of Fish, Plant and Gastronomy Technology, Faculty of Food Science and Fisheries, West Pomeranian University of Technology in Szczecin, 70-310 Szczecin, Poland; 2Department of Commodity Science, Quality Assessment, Process Engineering and Human Nutrition, Faculty of Food Science and Fisheries, West Pomeranian University of Technology in Szczecin, 70-310 Szczecin, Poland; 3Department of Plant Product Technology and Nutrition Hygiene, Faculty of Food Technology, University of Agriculture in Krakow, 30-149 Krakow, Poland

**Keywords:** oyster mushroom, antioxidant properties, common carp, fish burgers

## Abstract

The growing interest in functional food makes looking for new possibilities of enriching products with health-promoting ingredients necessary. One raw material with a very high potential for the food industry is the oyster mushroom (*Pleurotus ostreatus*), which has a strong antioxidant, antiviral, and anticancer effect. Carp meat (*Cyprinus carpio*) also has beneficial properties. It is rich in easily digestible protein, vitamins, minerals, and omega-3 fatty acids. This study aimed to evaluate the effect of oyster mushroom addition on the quality of carp burgers, with particular emphasis on the antioxidant effect. The scientific literature produced so far has not focused on the synergy between oyster mushrooms and carp meat. The addition of oyster mushrooms contributed to the increase in antioxidant properties and sensory attractiveness of burgers. The fat content in the finished product was reduced, and the degree of their oxidation was also reduced. The obtained results will contribute to the creation of innovative food products that meet the expectations of consumers looking for healthy food.

## 1. Introduction

The antioxidant activity and health-promoting properties of mushrooms are currently the subject of many studies. Their main goal is to determine the possibility of using edible mushrooms in food products in order to increase their nutritional value and quality. One raw material with high biological activity, including antioxidant potential, is the oyster mushroom [*Pleurotus ostreatus* (Jacq.) P. Kumm]. It is a mushroom belonging to the phylum Basidiomycota, order Agaricales. Its fruiting bodies can be obtained from a natural state (this mushroom inhabits dead trunks and logs of deciduous trees) or grown on specially composed but cheap substrates [1]. The oyster mushroom is edible and especially valued for its delicate taste. It is a source of protein [1,2,3], mineral salts, including magnesium [1,2,4], vitamins A, E, and D [1], and polysaccharides that can be found in cell walls [1,2,3,4,5,6]. However, the main component of fresh fruiting bodies is water, which can be about 94.8% of their weight [1,3]. In recent years, interest in oyster mushrooms as a raw material that can enrich food products with ingredients that give them new or improved properties has increased.

Carp meat (*Cyprinus carpio*) also has valuable health-promoting properties. It is a good source of protein, n-3 unsaturated fatty acids, vitamins, and minerals [7]. Although carp is one of the commercially important fish species farmed in Polish aquaculture [8] and most commonly eaten (about 0.50 kg per capita per year in Poland) among the domestic farm-raised freshwater species, freshwater fish consumption is still significantly low [9]. Moreover, it is popular only during the Christmas period. Many consumers declare they do not accept the specific taste and smell/flavor of carp meat. Moreover, carp is a very bony fish, which is a huge inconvenience in its consumption. There are preferred species of marine fish richer in meat with a lower share of pin bones. There is a possibility of avoiding bones in products by using mechanically separated fish meat. The efficiency of obtaining meat is high (level about 80%) and could be used to prepare fish burgers or paté of high nutritional value. However, including unacceptable taste according to some consumers, there is a possibility of forming/enhancing the sensory quality of carp burgers using mushroom addition. Alternatively, individuals increasingly opt for processed fish products like pre-packaged fish meals [10].

One of the ways to increase carp consumption throughout the year may be to develop new recipes and the possibility of combining the fish with other raw materials, thus obtaining innovative products with attractive sensory features and new nutritional and bioactive values. Carp is very well suited as a main course but also as a basic ingredient in processed products, such as burgers, which can increase the range of carp products. Carp burgers can also find buyers among young people and a group of consumers called flexitarians, who do not want to give up meat consumption but also want to consume plant products. In order to modify their taste, textural, and nutritional properties in this direction, steamed oyster mushroom fruiting bodies were added.

Mushrooms are characterized by being low in calories and fat while being abundant in proteins, dietary fiber (including chitin, hemicellulose, mannans, and glucans), and minerals [11,12]. Their chemical composition encompasses various bioactive substances, such as polysaccharides, biologically active proteins like enzymes and lectins, ergothioneine (amino acid), terpenoids, sterols, antioxidants, and vitamins like thiamine, riboflavin, ascorbic acid, niacin, and tocopherols. These components contribute to enhanced immune function and offer various beneficial effects, including anti-tumor, antimicrobial, anti-inflammatory, hypoglycemic, hypocholesterolemic, and antioxidant properties [13].

Although there are some studies of mushroom application in beef burgers, pork sausages, meatballs, and beef or chicken pate to reduce fat content [14,15,16], there were no publications about the application of mushrooms, especially oyster mushrooms, to enhance the quality of carp meat products. Our research focused on the possibility of an innovative combination of oyster mushroom, as a mushroom with high antioxidant activity, with carp meat in order to create an alternative product to traditional burgers. The aim of these studies was to determine the antioxidant potential of oyster mushrooms and the effect of this raw material on the basic composition and oxidative, lipid, and sensory properties of carp burgers.

## 2. Results and Discussions

### 2.1. Raw Materials

#### 2.1.1. Basic Characteristic of Oyster Mushroom and Carp Meat

The process of steaming the mushrooms caused a significant decrease in the water and ash content in the tissues. Other changes in the content of ingredients were not significant. The water content in carp meat was significantly lower than in fresh and steamed mushrooms. However, this raw material showed a higher content of protein, approx. 19.6%, and fat, approx. 4.9%, in comparison with both forms of mushrooms (Table 1). Muyanja et al. [17] described the effect of various forms of processing and drying of fruiting bodies of *Pleurotus ostreatus* on the basic composition of the obtained raw material. They showed significant differences in the content of water, protein, fats, ash, and carbohydrates depending on the form of drying (sun in the open air or in the oven), the use of pre-heat treatment (blanching), and the salting time. Water and protein content were always subject to the greatest changes; they always decreased. The steaming method used in our research is a form of high-temperature impact on the raw material [18]. This type of technological process causes damage to the cytoplasmic membranes in mushroom cells, which promotes leakage of cell fluid. The result is a decrease in the water content of the tested raw material.

#### 2.1.2. Antioxidant Properties and Total Polyphenols Content of Oyster Mushroom and Carp Meat

Carp meat was characterized by low antioxidant properties determined against free radicals ABTS, DPPH, and ferric-reducing ability (FRAP) (Table 2). Fresh oyster mushrooms showed antioxidant properties against (ABTS +) that were more than twice as strong as carp meat (4.68 µmol TE/g and 1.98 µmol TE/g, respectively). Ferric-reducing antioxidant power of mushrooms was 50% higher than the FRAP of carp meat (29.8 µmol TE/g and 18.9 µmol TE/g, respectively). The higher antioxidant properties of oyster mushrooms may be related to the approximately 60% higher total phenolic compounds (TPC) content in mushrooms than in carp meat. Yang et al. [19] found that oyster mushrooms showed the highest antioxidant properties among six species of commercial mushrooms regardless of the determination method (FRAP, DPPH scavenging ability, FCA, and others). Antioxidant properties of mushrooms result mainly from the high content of polyphenols [20] but also from the presence of polysaccharides, which show DPPH radicals scavenging ability [21]. Steaming caused a decrease in the TPC content due to leakage of cell fluid during the process, while the antioxidant activity [ABTS, FRAP, and DPPH] of steamed oyster mushrooms was not significantly different from that of raw oyster mushrooms. The ferrous chelating ability (FCA) increased after steaming but was still about four times lower than the chelating capacity of meat. The content of total polyphenols (TPC) in oyster mushrooms in our study was higher than in the study of Florczyk et al. [22] and González-Palma et al. [23], as was the ability to reduce iron ions of FRAP [23,24]. On the other hand, the antioxidant activity of oyster mushroom fruiting bodies against both the ABTS and DPPH radicals in our study was lower than in the studies of Lam and Okello [24], as was the antioxidant activity of carp meat against the ABTS and DPPH radicals [25]. 

Polyphenolic compounds contained in oyster mushrooms are characterized by many health-promoting properties, including antioxidant, anticancer, anti-inflammatory, and antimicrobial properties [11,26,27]. However, there is still a lack of available knowledge indicating the typical polyphenolic composition in this species of mushrooms [28]. So far, studies have identified over a dozen phenolic compounds in oyster mushrooms, mainly hydroxybenzoic acid, sinapic acid, ferulic acid, coumaric acid, protocatechuic acid, vanillic acid, caffeic acid, gallic acid, chlorogenic acid, homogentisic acid, dihydroxybenzoic acid, hydroxyphenylacetic acid, cinnamic acid, catechin, and gallocatechin [11,27,28,29,30]. However, many of them were identified in some studies but were not confirmed in others [11,27,28]. Qualitative and quantitative differences between studies may be the result of different methods of cultivation, preparation, and extraction, as well as geographical variability [11]. In this study, several standards were used to identify polyphenols in oyster mushrooms by means of liquid chromatography (gallic acid, chlorogenic acid, caffeic acid, ferulic acid, coumaric acid, catechin, epicatechin, rutin, kaempferol, apigenin, and quercetin), but only two compounds were identified, i.e., epicatechin and gallic acid (Table 3, Figure 1). In 100 g of fresh mushrooms, the average content of epicatechins and gallic acid was 106.6 mg and 34.2 mg, respectively.

The lower content of gallic acid in oyster mushrooms was determined by Alam et al. [30]—36 µg/g (3.6 mg/100 g). Differences in the content of this acid may result from analytical differences, including different extraction methods used, as well as different environmental conditions for the growth of both compared mushrooms. Gallic acid, next to *p*-hydroxybenzoic acid and protocatechuic acid, shows high biological activity and effectively protects against free radicals [11].

On the other hand, in the case of epicatechins, despite being quite common in various species of mushrooms [31], catechin was identified in published studies more often than its isomer (epicatechin) [32].

A significant effect of steaming on the content of polyphenols in oyster mushrooms was found. The heat treatment process resulted in a decrease in the content of epicatechins and gallic acid (39% and 80%, respectively) compared to their content in fresh mushrooms before processing (Table 3). The losses of both compounds most likely result from the negative impact of elevated temperature on their molecules. Thermal degradation of epicatechin has also been observed in other studies [33].

In general, the effect of thermal processing, including steaming, on polyphenols in food products is very diverse. Some studies show losses of polyphenols as a result of evaporation, and some show a positive effect of high temperature on their content [34]. Polyphenols can oxidize and decompose under the influence of elevated temperatures [35], as well as a result of the activity of endogenous enzymes [36]; on the other hand, high temperatures, due to the disruption of the structure of cell walls, may release more active forms—aglycones—from the glycoside forms [34].

#### 2.1.3. Determination of Lipid Quality Parameters of Oyster Mushroom and Carp Meat

Based on the results presented in Table 4, it can be seen that the content of the peroxide value (PV), anisidine value (AsV), and hydrolytic changes (AV) is the highest for meat compared to raw and steam-treated oyster mushrooms.

The PV for fresh mushrooms was 4.116 ± 0.044 meqO_2_/kg of fat; after steam treatment, it increased to 5.020 ± 0.115 meqO_2_/kg of fat. The PV for meat was 6.433 ± 0.062 meqO_2_/kg of fat—this may indicate greater lipid peroxidation processes in meat compared to mushrooms. These changes may be the result of the presence of a greater amount of easily oxidizable fatty acids in meat compared to mushrooms [36]. The anisidine value and hydrolytic changes are also the highest for meat, which indicates a greater amount of free fatty acids produced as a result of lipid hydrolysis. Comparing steamed and fresh oyster mushrooms, the AsV and AV are also higher in the case of steamed mushrooms by 19% and 27%, respectively. The increase in lipid quality parameters in steamed oyster mushrooms can be attributed to thermal degradation and, thus, changes in lipids [37].

### 2.2. Fish Burgers

#### 2.2.1. Basic Characteristic of Carp Burgers

Five types of fish burgers were prepared, differing in content of oyster mushrooms. The control sample (P0) was the sample without the addition; P5 contained a 5% addition of oyster mushrooms; P10 contained a 10% addition of oyster mushrooms; P15 contained a 15% addition of oyster mushrooms; and P20 contained a 20% addition of oyster mushrooms. Fish burgers were weighed before and after thermal treatment. Table 5 shows the losses resulting from frying. It can be seen that the addition of oyster mushrooms contributed to smaller weight losses. Weight loss in P0 was 9.31%, while the addition of 20% of oyster mushrooms to the fish burger reduced the loss to 8.88%. These changes were not significant.

The basic composition of mushrooms in our research was very similar to the composition of fruiting bodies of the genus Pleurotus shown in other studies [11,38,39].

The addition of mushrooms to burgers significantly affected the content of water, protein, and fat (Table 6). The product with the highest content of mushrooms (P20) had the highest water content. It also has the lowest protein and fat content. The addition of oyster mushrooms to burgers did not affect the ash content. The lack of changes in the ash content in burgers is due to the fact that oyster mushrooms are rich in structural polysaccharides, especially fiber and glucans, which affect the amount of dry matter in the mushroom [12]. On average, oyster mushrooms contain over a dozen percent of dry matter, which consists mainly of carbohydrates, including mainly complex ones (37–48%), fiber (13–24%), protein (20–25%), lipids (4–5%), and minerals (8–13%) [13]. Therefore, by replacing the fish meat in the burger with mushrooms, no changes in the ash content are noticed. The effect of the addition of *Pleurotus* mushrooms on the basic composition of the products has also been demonstrated in other studies. Most often, a significant increase in the product’s water content, an increase in the content of protein, fiber, and ash, and a decrease in the content of fat and carbohydrates were observed. However, the size of the differences depended on the type of product and the basic composition of other raw materials used in its preparation [14,16,40,41,42].

#### 2.2.2. Antioxidant Properties and Total Polyphenols Content of Carp Burgers

The addition of mushrooms to burgers resulted in an increase in the content of total phenolic compounds, but no significant differences were found between the samples with the addition of mushrooms of 5 and 10%, as well as 15 and 20% (Table 7). The antioxidant capacity against ABTS increased with the share of oyster mushrooms in burgers, although a significant increase was noticed with the addition of the mushroom at the level of 10% or more. In turn, the ferric-reducing power increased significantly after adding oyster mushrooms but did not differ between burgers with different amounts of oyster mushrooms. The DPPH radicals scavenging ability significantly increased only with the use of 20% oyster mushroom. The DPPH scavenging ability is affected by polyphenols and the contents of amino acids and small peptides, especially compounds containing aromatic groups [43]. Although the content of non-protein nitrogen and amino nitrogen amount did not differ significantly, despite increasing the share of oyster mushrooms in burgers, the amount of α-amino nitrogen increased, reaching a value twice as high as in the sample with the addition of 20% oyster mushroom than in the control sample. Turfan et al. [20] found a relatively high content of free amino acids in extracts of dried oyster mushrooms. The results of determining protein hydrolysis products using the Lowry method differed between the samples; however, it should be noted that both aromatic amino acids and polyphenols react with the Folin-Ciocalteau reagent [44]. Therefore, changes in PHP-peptides and PHP-Tyrosine amounts in burgers may result from the participation of amino acids, peptides, and polyphenols. Thus, the amount of PHP seems higher in mushrooms than in meat. The chelating of iron (II) ions ability decreased with the increasing share of oyster mushrooms (1.46 µmol EDTA/g in the control sample and 1.2 µmol EDTA/g in P20) due to them having a lower chelating ability than meat. However, the FCA of burgers was about 35% lower than that of carp meat.

The addition of oyster mushrooms enriched the burgers with polyphenolic compounds. With the increasing amount of mushroom addition to burgers, the content of epicatechins and gallic acid increased significantly (Table 8). About 12.9 mg of epicatechins and 1.4 mg of gallic acid were found in 100 g of burgers with 20% oyster mushrooms (Table 8, Figure 2).

#### 2.2.3. Determination of Lipid Quality Parameters of Carp Burgers

Oxidative changes

Figure 3 shows the changes in primary oxidation products expressed as PV. Comparing the effect of oyster mushroom addition on the degree of oxidation, it was found that the addition of oyster mushrooms to fish burgers reduced the content of peroxides. With the addition of oyster mushroom, the peroxide value decreased to 7.897 ± 0.009 meqO_2_/kg of fat in sample P5 and 5.949 ± 0.016 meqO_2_/kg of fat in sample P20. The high-fat content of the control sample is one of the reasons for its lower oxidative stability. Oxidation reactions in fried burgers occur at various stages of processing (grinding, mixing, and cooking) and storage due to various types of oxidizing agents in the muscle tissue [45]. The results were consistent with the study by Patinho et al. [15], where the addition of Agaricus bisporus was investigated as a partial fat replacement in beef burgers. They also showed that the addition of mushrooms has a beneficial effect on reducing and slowing down changes in the primary products of fat oxidation.

The number of secondary oxidation products, expressed as the anisidine value (AsV), decreased with the increasing amount of oyster mushrooms in the product (Figure 4). The control sample was 9.19 ± 0.02, and the P20 sample was 6.58 ± 0.05. For the P10, P15, and P20 samples, there was a significant (*p* < 0.05) reduction when compared to the control sample.

The use of the test expressed in AsV allows the determination of the TOTOX index, which allows full visualization of the level of oxidation (Table 9). This index was used in relation to the Codex Alimentarius [46] fish oil oxidation limits, which is ≤26. Comparing the obtained results, it was found that with a greater addition of oyster mushrooms, the TOTOX value decreases. In addition, it is worth noting that the TOTOX value does not exceed 26 in any case. Its highest value was observed in the control sample (25.35).

Hydrolytic changes

The quality assessment of fish oils involves the determination of acid value, which can serve as an indicator not only for fish but also for other food products, such as fish burgers. In the case of hydrolytic changes, similar dependencies to those observed in oxidative changes can also be noticed. With a greater source of oyster mushrooms, the resulting acid value decreases and is significantly different from the control and P5 (Figure 5). In the control sample, the content of acid value was 1.75 ± 0.03 mg KOH/g fat, while in the sample from the main component of the oyster mushrooms (P20), the content was 1.06 ± 0.05 mg KOH/g fat. Similar results were observed by Nayak et al. [47], where a decrease in the percentage of free fatty acids after the addition of mushrooms to patties prepared with sutchi was proved.

#### 2.2.4. Sensory Analysis of Carp Burgers

Table 10 presents the results of the sensory evaluation of the tested burgers. P0, according to the evaluators, was characterized by a good general appearance (4 points) and did not differ significantly from burgers with a 20% addition of oyster mushrooms. In the remaining products, the increasing share of the mushrooms from 5% to 15% improved their general appearance. Among them, the highest score (5 points) was awarded to the P15 burgers, which did not differ significantly from the P10 burgers (4.86 points). In terms of color, burgers with 15% oyster mushroom were rated at the highest 5 points and the lowest at 3.86 points—the samples in which its share was 5% and 20%. For comparison, the color of the control sample was rated at 4 points. The addition of oyster mushrooms favorably shaped the taste of burgers in the concentration range from 5% to 15%. The best taste was considered to be the P15 sample, to which the evaluators gave 4.86 points, and the P10 sample was not significantly different from it (4.71 points). The taste of burgers with 20% oyster mushroom was rated the lowest at 3.86 points. The taste of sample P0 was rated as good (4.14 points). In the evaluation of the smell, the highest score of 5 points was given to samples P10 and P15. In other burgers with the addition of oyster mushroom 5% and 20%, the smell was also very pleasant—4.57 points and 4.43 points. Similarly, in the control sample (without the addition of oyster mushroom), the smell was rated at 4.28 points, which was typical for carp burgers. In terms of texture, adding oyster mushrooms in the concentration ranges from 10% to 15% profiled the evaluated feature most favorably—this corresponded to the following scores of 4.71 for the P10 and 4.86 for the P15. Regarding texture, the control variant (without the addition of the mushroom) scored 3.86 points, and with 20% of oyster mushrooms—3.85 points. Among the evaluated burgers, the addition of oyster mushrooms at a concentration of 15% profiled their sensory characteristics in the best way. According to panelists, products with the participation of mushrooms showed the highest acceptability in this respect. In burgers with 10% oyster mushroom, their acceptability was also high (4.86 points). A similar effect on sensory characteristics with the addition of mushrooms in the range of 10% and 15% was obtained in beef burgers by Patinho et al. [16]. In addition, in our research, in which oyster mushrooms accounted for 20%, the acceptability of the product was rated the lowest at 3.57 points, and it differed significantly from the others.

On the basis of a detailed assessment of the taste of fried burgers, it was shown that the degree of perceptibility of its individual characteristics changed depending on the addition of oyster mushrooms (Figure 6). In the control sample (P0), without the participation of the mushroom, the taste of the meat was significantly (*p* < 0.05) dominant, followed by the spices and greasy taste, and the spice flavor was moderately dominant. The evaluation team also tasted a slightly salty and fishy taste. In the case of samples with the addition of oyster mushroom, the perceptibility of the taste of the mushroom significantly (*p* < 0.05) increased with the increasing in its concentration, from very weak (P5) to clear (P20). The taste of meat in these samples, similarly to salty, was moderate and slightly above moderate. The obtained results did not differ statistically (*p* < 0.05). The research of Patinho et al. [16] also showed that adding mushrooms (10% and 15%) improved the taste of burgers and emphasized their saltiness. According to Dermika et al. [48], this increase in the perceptible taste of salty burgers was the result of the compounds contained in mushrooms: glutamic acid, aspartic acid, and 5′-nucleotides responsible for their umami taste. As confirmed by Jo Feeney et al. [49], the addition of mushrooms also increased the perceptibility of spices and meat, which is confirmed by our study. In sample P0, compared to all the other samples, the taste of spices was significantly (*p* < 0.05) stronger, and the taste of fish was significantly (*p* < 0.05) weaker. With a 20% share of oyster mushrooms, the burgers had a significant (*p* < 0.05) additional taste of slight bitterness. All evaluated samples were free of rancid taste. According to the evaluators, the addition of mushrooms up to 15% significantly (*p* < 0.05) and positively influenced the profiling of the taste of carp burgers.

The profile assessment of the aroma of fried burgers with the addition of oyster mushroom showed that three distinguishing features, such as mushroom, meat, and spices, had the main share in shaping it (Figure 7). The intensity of detecting the mushroom smell notes increased in the samples with its addition from very weak (P5) to slightly above moderate (P20). Even the addition of 10% of mushrooms had a significant (*p* < 0.05) effect on its perceptibility. According to Misharin et al. [50], aliphatic alcohols and ketones are responsible for the pleasant smell of oyster mushrooms: 1-octen-3-ol, 2-octen-1-ol, 3-octanol, 1-octanol, 1-octen-3-one, and 3-octanone. Thanks to them, carp burgers acquire the desired aroma, which improves their quality. In the case of the meat and spices odor discriminant, their perceptibility was at a similar level, with a score between 3 and slightly higher. In sample P0, the evaluators detected an above-moderate odor of meat and a slight odor of spices. However, the perceptibility of this smell was not significantly different (*p* < 0.05) from its intensity in other samples, whereas the fishy smell in this sample was very weak, and the participation of the mushroom was almost nonexistent. According to the evaluators, all samples were free of sour and rancid odors, which indicates the high freshness of the components used in the technology.

Evaluating the texture of the finished carp burgers, it was noted that the degree of juiciness sensed in them was pronounced, similar, and insignificant (*p* < 0.05) for all samples, regardless of the addition of oyster mushroom (Figure 8). With the increase in the share of the mushrooms in the samples, their greater tenderness and softness were felt. The samples containing 15 and 20% of the mushrooms differ significantly (*p* < 0.05) in terms of tenderness and juiciness from the remaining samples. Fibrousness was not felt in them, apart from a very weak one in the control sample (without the addition of oyster mushrooms). According to the panelists’ assessment, the use of mushrooms in the amounts of 10% and 15% most favorably profiled the texture of carp burgers. In the case of the sample with the highest amount of mushrooms—20%, the texture was too fragile, too juicy, and clearly softer and slightly deviated from the right one. The control sample (without the addition of oyster mushrooms) was less soft and fragile, which resulted in its slightly lower acceptability than in the tests with oyster mushrooms (P10 and P15). The obtained texture results are confirmed by the research of Afshari et al. [51], Moghtadaei et al. [52], and Patinho et al. [16]. According to the authors, the addition of mushrooms reduced the hardness of the burger texture and gave it a delicate softness. As stated by Alakali et al. [53], mushrooms, due to the high capacity of water retention in the product, increased their juiciness and reduced the degree of shrinkage, which had a positive effect on maintaining the proper shape.

#### 2.2.5. Objective Method of Color Parameters of Carp Burgers

Giving the fish burgers the right color, shaped by the addition of oyster mushrooms and the components used in their technology, is very important. Their desired color will always indicate high quality, attract attention, and encourage consumption. Therefore, creating it in the right direction using the measurement of objective color parameters is highly recommended.

Table 11 presents the results of the effect of oyster mushroom addition on the values of objective color parameters and the control sample (without mushroom addition). P0 fried burgers showed a lower color lightness (L* = 54.35) than the samples with the addition of oyster mushrooms. The samples with mushroom participation in the amounts of 5% and 10% did not differ significantly (*p* < 0.05) in terms of the L* parameter, unlike the others. The P20 burgers were characterized by the highest color lightness of L* = 57.49 among those evaluated. The increasing share of oyster mushrooms in the evaluated burgers resulted in an increase in color lightness by 2% (P5) to approximately 6% (P20) compared to P0. With regard to the redness of the color, the P0 and P5 samples did not differ significantly (*p* < 0.05), for which the parameter a* was 10.38 and 10.06, respectively. The variants with 10, 15, and 20% percent addition of oyster mushrooms differed significantly in redness (*p* < 0.05). Generally, the increased addition of oyster mushrooms in the evaluated burgers resulted in a reduction in the redness of the color by 3% (P5) to 22% (P20) compared to P0. The yellowness of the color of the samples, similar to their redness, decreased with the increasing addition of oyster mushrooms. P0 carp burgers (without the addition of oyster mushroom) were characterized by the highest b* parameter (17.50), while the lowest b* parameter was noticed for sample P20 (14.51). The samples with the addition of oyster mushrooms in the amount of 5 and 10% did not differ significantly (*p* < 0.05) in yellowness (b* = 16.82 and b* = 16.21, respectively) compared to the others. In the evaluated burgers, with the increase in the addition of oyster mushrooms, the yellowness of the color decreased by 4% (P5) to 17% (P25) in relation to P0. Patinho et al. [16] showed that the addition of mushrooms to beef burgers in amounts of 5% to 15% caused an inverse relationship—a decrease in the lightness of the L* parameter and an increase in the a* and b* parameters. For comparison, Wong et al. [54] determined the L* and a* color parameters in minced beef cutlets with a 10% mushroom addition. The authors found that the results obtained were similar to the control variant (without the addition of mushroom); however, they differed from it in that the parameter b* increased significantly (*p* < 0.05). As confirmed by Kurt and Gençcelep [55], changes in the color components of burgers depend not only on the specificity of the raw material but also on the addition of components (the amount of mushroom, spices, and fat) that affect their overall color. All the evaluated samples differed significantly in color saturation; the control variant showed the highest value (20.74). The increasing share of oyster mushrooms in the evaluated samples caused a decrease in color saturation from 7% (P5) to 18% (P20) in relation to P0. The determined values from 60.23 (P0) to 54.12 (P20) for the color tone of fried burgers differed significantly at the level of *p* < 0.05. As more oyster mushrooms were incorporated into the samples, the H* parameter value decreased within the range of 2% to 10%. Regarding the ΔE parameter, the most substantial and statistically significant changes (4.08) were observed in the P20 carp burgers. The increasing addition of oyster mushrooms in burgers decreased the H* parameter by 2% (P5) to 10% (P20) compared to the control sample.

#### 2.2.6. Comparative Analysis

Principal component analysis (PCA) was conducted to gain a deeper understanding of the relationships and associations among the various components. PCA analysis was performed for antioxidant activity, fat value, and sensory analysis and distribution of five samples of fish burgers with different oyster mushroom additions. PCA demonstrated that the initial two components explain 96.69% of the overall variance, as illustrated in Figure 9A. The strongest correlation in the first factor was found between ASV (0.99) and FCA (−0.97), as well as between ABTS (−0.98) and TPC (−0.96). In addition, in the second factor, a very strong correlation can be distinguished between taste (0.96) and general desirability (0.96), which showed that taste was the most important parameter for sensory evaluation. The weakest correlation was found for FRAP (−0.69) in relation to other factors. The analysis permits the allocation of the samples into the quadrants of the two-factor case coordinate plot, as shown in Figure 9B. The sample comprising the one-piece set in quadrant II is P15, which had the best sensory ratings and FRAP activity. Quarter III included the P20 sample, which was strongly related to the remaining antioxidant values and the weakest to the sensory evaluation parameters. In quarter IV, a separate group was formed by the P0 sample, which had the lowest antioxidant activity and was characterized by the highest activity of fat value. However, it was similar to the P5 sample in quarter I, which was also characterized by low antioxidant values and an increased content of fat value.

## 3. Materials and Methods

### 3.1. Materials

The main raw material for the production of burgers was mechanically separated flesh from common carp (MSFC). The common carp (*Cyprinus carpio*, L.) came from a local fish farmer in the West Pomeranian Voivodeship. Carp was transported in ice to the laboratory in boxes made of expanded polystyrene (Atlantic Styro A/S, Morawica, Poland) with a capacity of 25 kg. After removing the ice, fish were beheaded, gutted, and washed. After washing in running water, carcasses were placed on a perforated tray with the spine up and left to leach out. Then, carcasses were passed through the drum separator (type NF 13DX Bibun, Fukuyama, Japan, diameter of hole 4.0 mm) and cleaned thoroughly in a screw separator (Steiner, type SUM 420, Bibun, Fukuyama, Japan, diameter of hole 2.0 mm). 

Fresh oyster mushrooms (*Pleurotus osteatus*) (G&G FUNGI, Salamony, Poland) were purchased from a local store and processed immediately after delivery to the laboratory. After cleaning, the mushrooms were steamed for 20 min (Philips electric steamer, HD9126/90, Philips, Warszawa, Poland) and, after cooling down, crushed with an immersion blender (Hand mixer, Hendi 250 VV, Hendi, Warszawa, Poland).

### 3.2. Preparation of Fish Burgers

Prior to the preparation of the samples, MSFC was stored in a cold store at 4 ± 1 °C.

The burgers were prepared as follows: comminuted carp meat (85%) was combined with addition of wheat flour (3.2%), egg powder rehydrated 30 min earlier (0.8% and 2.4% of water), tomato concentrate (3.2%), seasonings as dried garlic (0.15%), black pepper (0.075%), herb pepper (0.15%), hot pepper (0.025%), salt (1.0%) and rapeseed oil (4.0%).

All ingredients were mixed together and formed by hand using a press. The resulting burger with a diameter of 8 cm, a height of 10 mm, and a weight of 75 g was fried in a pan in hot rapeseed oil (170 ± 2 °C) to 75 ± 2 °C inside the burger (5 ± 1 min of frying). After draining excess fat, the burgers were weighed and then weighed again after cooling down (reaching room temperature 20 ± 2 °C). The burgers were golden brown in color. Sensory evaluation was carried out on still-warm burgers (62 ± 2 °C), and texture analyses, color determinations, and extracts for other chemical analyses were performed directly on the cooled samples.

The control sample was the burger without the oyster mushroom addition. In the case of the version with the addition of oyster mushrooms, part of the meat was replaced with an appropriate amount of steamed mushrooms, which accounted for 5 to 20% (Table 12). The samples were marked with symbols indicating the participation of mushrooms (from P0—no participation to P20—20% share of mushrooms).

### 3.3. Methods

Five versions of burgers were prepared in three separate batches, each yielding twelve burgers. This approach allowed for three replications for each burger version. Additionally, the raw materials (carp meat and mushrooms) underwent testing five times. This multiple testing was conducted due to their high susceptibility to changes in storage conditions and to eliminate potential influences associated with a single batch.

All analyses were presented in triplicate and were converted to wet weight.

#### 3.3.1. Preparation of Extracts for Analyses

Trichloroacetic acid (TCA) extracts. The content of non-protein nitrogen was determined using prepared TCA extracts. Comminuted burgers were homogenized with 5% (*v*/*v*) TCA (5 g sample/100 mL 5% TCA *w*/*v*) for 1 min, then left for 5 min, and after 5 min, homogenization was repeated. After that, the homogenate was left for 30 min. Afterward, the extract was filtered and collected in a dark glass bottle.

Methanolic extracts. Antioxidant activity was determined on prepared methanol extractions. Comminuted burgers were homogenized with 80% (*v*/*v*) methanol (10 g sample/100 mL 80% MeOH *w*/*v*). Afterward, the extract was filtered and collected in a dark glass bottle.

Chloroform-methanol extract. Lipids were extracted using the Bligh and Dyer [56] method. Chloroform and methanol were added to the sample in a ratio of 1:1 and then homogenized. The extract was filtered under pressure and poured into a separatory funnel. After adding water, two layers were obtained, which were separated. Quantification results were expressed as grams of lipid per 100 g of muscle tissue.

#### 3.3.2. Analysis of General Composition

The dry matter, water content, total nitrogen (analyzed using the Kjeldahl method with Kjeltec KT 200 apparatus from Labtec Line, FOSS, Warszawa, Poland), fat content (determined by the Soxhlet method), and ash content (measured using the dry mineralization method) were assessed following the protocols outlined by the Association of Official Analytical Chemists (AOAC) [57].

The content of non-protein nitrogen was determined in the Kjeldahl method [57]. Protein hydrolysis products (PHP): tyrosine [PHP-tyrosine] and peptides [PHP-peptides] were determined with a modified Lowry method [44]. Alpha-amine nitrogen was determined with the Pope-Stevens method [58], and amine nitrogen was determined with the Sorensen method [59]. 

Lipids were extracted by cyclohexane and propan-2-ol and transferred to the cyclohexane phase by the addition of water. Phase separation was performed by centrifugation. Gravimetric fat was determined after the separation of the cyclohexane layer and evaporation to dryness [60].

#### 3.3.3. Total Phenolic Compounds Content

Determining the total phenolic compound content (TPC) was conducted using methanolic extracts per the method described by Turkmen et al. [61]. In brief, 1 mL of the diluted extract was mixed with 5 mL of 10% Folin–Ciocalteau reagent, followed by the addition of 4 mL of 7.5% Na_2_CO_3_ after a 5 min interval. Subsequently, the mixture was left to incubate in the dark for 2 h, and the absorbance was measured at 750 nm using a Helios Gamma spectrophotometer (Thermo Spectronic, Horsham, UK). The results were expressed as milligrams of gallic acid equivalent per g (mg GAE/g).

#### 3.3.4. Antioxidant Properties

Antioxidant activity of meat, mushroom, and fish burgers was determined in methanolic extracts as: 

Trolox equivalent antioxidant capacity was determined against ABTS (2,2′-Azino-bis(3-ethylbenzothiazoline-6-sulfonic acid) diammonium salt) cation radical according to Re et al. [62]. In this procedure, 4 mL of the ABTS solution (7 µmol solution activated by K_2_S_2_O_8_ and adjusted to an absorbance of 0.700 ± 0.020 just before analysis) was mixed with 40 μL of the tested extract. This mixture was then left in the dark for 30 min. After this incubation period, the absorbance was measured at 734 nm (Helios Gamma spectrophotometer, Thermo Spectronic, Horsham, UK). Total antioxidant activity was expressed as µmol Trolox/g based on the standard curve.

Ferric-reducing antioxidant power (FRAP) was determined according to Benzie and Strain [63]. Briefly, 3 mL of a working solution (comprising 1 part 0.01 mol TPTZ (2,4,6-Tri(2-pyridyl)-s-triazine) in 0.04 mol HCL, 1 part 0.02 mol FeCl_3_, and 10 parts 0.3 mol acetate buffer pH 3.6), previously heated to 37 °C for 30 min, was added to 100 μL of the sample. The samples were incubated for 30 min at room temperature in the dark, and then the absorbance at 593 nm was measured (Helios Gamma spectrophotometer, Thermo Spectronic, Horsham, UK). Antioxidant activity was expressed as Trolox equivalents [µmol Trolox/g] using a standard curve.

DPPH radical scavenging activity was determined according to Brandt-Williams et al. [64]. In brief, 4 mL of methanolic extract was combined with 1 mL of 0.2 μmol DPPH˙ in methanol. These samples were then shaken and left in the dark for 30 min, after which the absorbance was measured at 517 nm using a Helios Gamma spectrophotometer (Thermo Spectronic, Horsham, UK). The ability to scavenge free radicals was expressed as μmol Trolox per gram, based on the reduction in absorbance relative to the control (%) using the Trolox standard curve.

Fe^2+^ ions chelating ability (FCA) was determined according to Khantaphant et al. [65]. Thereafter, 0.1 mL of 2 mmol/L FeCl_2_ and 0.2 mL of 5 mmol/L ferrozine were added. The mixture was allowed to react for 20 min at room temperature. The absorbance was then measured at 562 nm (Helios Gamma spectrophotometer, Thermo Spectronic, Horsham, UK). Ferrous chelating ability was expressed as μmol EDTA/g based on the standard curve.

#### 3.3.5. Extraction of Active Compounds (polyphenols) from Fish Burgers and Oyster Mushrooms and Identification by Liquid Chromatography (HPLC) 

The Method was validated with respect to linearity, LOD (limit of detection), LOQ (limit of quantification), recoveries, and precision according to AOAC guidelines [66]. Fish burgers and mushrooms were analyzed for gallic acid, chlorogenic acid, caffeic acid, ferulic acid, coumaric acid, catechin, epicatechin, rutin, kaempferol, apigenin, and quercetin using the developed method. All the analytes exhibited good linearity (r) over the range tested, with correlation coefficients ranging from 0.9959 to 0.9997. Precision and accuracy were evaluated using recovery tests (*n* = 6), adding known amounts of the standard solution to an extract. The percentage recoveries were calculated, which were always higher than 97.2%. Relative standard deviations (%RSD) of all compounds were less than 10%.

Active components from fish burgers were extracted using two different solvents, hexane, and 70% methanol, in an ultrasonic bath following the method described by Hrebień-Filisińska and Bartkowiak [67]. On the other hand, flavonoids and polyphenolic acids from oyster mushrooms were extracted using 70% methanol in an ultrasonic bath for a duration of 10 min. To analyze the individual active compounds in these extracts, liquid chromatography (HPLC—Agilent 1260 Infinity II liquid chromatograph (Agilent Technologies, Inc., Santa Clara, CA, USA) equipped with a PDA detector was utilized). The separation was conducted on a Nucleosil 120-5 C18 reverse phase column measuring 250 × 4.6 mm at room temperature. The mobile phase consisted of acetonitrile (referred to as solvent A) and water with 5% acetic acid (referred to as solvent B). The flow rate was maintained at 0.5 mL/min. The gradient program involved starting with 15% A and 85% B for the first 12 min, followed by a linear change to 0% A and 100% B at 30 min. Subsequently, the gradient was altered to 85% A and 15% B at 50 min and maintained for an additional 10 min. The total analysis time for each sample was 60 min. An injection volume of 20 µL was used, and the peaks were monitored at multiple wavelengths, specifically 270, 280, 290, and 325 nm. Prior to injection, the samples were filtered through a 0.45 µm membrane filter (PTFE, hydrophilic, pureland 0.45 µm, Chemland, Warsaw, Poland). For reference standards of polyphenolic compounds with purity exceeding 98%, including phenolic acids (gallic acid, chlorogenic acid, caffeic acid, ferulic acid, coumaric acid) and flavonoids (catechin, epicatechin, rutin, kaempferol, apigenin, quercetin), these were dissolved in methanol, filtered through a 0.45 μm filter, and then promptly injected into the HPLC column for analysis. Identifications were based on a comparison of retention times with a standard at different wavelengths. The contents of epicatechin (λ = 280 nm, retention time—8.7 min) and gallic acid (λ = 270 nm, retention time—6 min) identified in the samples were calculated from the standard curve.

#### 3.3.6. Determination of Lipid Quality Parameters of Raw Materials and Carp Meat

Peroxide value (PV) was determined in the lipid extract by peroxide reduction with ferric thiocyanate, according to Pietrzyk [68], based on the oxidation of ferrous salt by hydroperoxides and the reaction of ferric salts with potassium isothiocyanate. The red ferric complexes formed were determined spectrophotometrically. Results were expressed as milliequivalents of oxygen per kilogram of lipids (meqO_2_/kg of lipids). 

Anisidine value (AsV) was determined in fish muscle according to the American Oil Chemists’ Society method (AOCS) [69], based on the reaction between α- and β-unsaturated aldehydes (primarily 2-alkenals) and *p*-anisidine reagent. AsV was expressed as 100 times the absorbance measured at 350 nm (Thermo Scientific, Waltham, USA, Helios Ga) in a 1 cm path length cuvette from a solution containing 10 mg of lipid in 1 mL of reaction medium. 

Acid value (AV) was determined by titration of 0.1 N KOH in methanol, according to the Polish Standard method [70]. The results are expressed in mg KOH/g fat.

The TOTOX index was calculated using the equation: (1)2×PV+AsV

#### 3.3.7. Sensory Analysis of Carp Burgers

In carp burgers with and without the addition of oyster mushrooms (control sample), a sensory assessment of appearance, color, taste, smell, and texture was carried out using a 5-point scale (1—poor, 2—insufficient, 3—sufficient, 4—good, and 5—very good) according to Baryłko-Pikielna and Matuszewska [71] with the participation of a 7-person team properly trained in terms of sensory sensitivity [72]. For a more accurate characterization of the taste and aroma characteristics of the products obtained, 11 flavor characteristics were assessed (fishy, meat, mushroom, spices, salty, sweet, bitter, sour, bland, greasy, rancid) and 11 identical odor indicators on a 6-point scale (0—imperceptible, 1—very weak, 2—weak, 3—moderate, 4—clear, and 5—very clear) according to Samotyja et al. [73]. In addition, the overall acceptability of the product was assessed on a 5-point hedonic scale in accordance with ISO 4121:2003 [74] (1—very undesirable, 2—undesirable, 3—neither undesirable nor desirable, 4—desirable, and 5—very desirable) and the intensity of perceptibility of texture characteristics (juicy, crispy, fibrous, soft, and fine) on a 6-point scale [71].

#### 3.3.8. Objective Method of Color Parameters of Carp Burgers

The color of fried burgers with and without the addition of steamed oyster mushrooms (control sample) was assessed by the objective method on the NH 310 apparatus (Shenzhen Technology Co., Ltd.; Shenzhen, China) after they were cut across. The following color parameters were analyzed: L* (lightness); a* (red/+) or (green/−) and b* (yellow/+) or (blue/−) in seven replicates were calculated. Moreover, the overall difference in the color of burgers (ΔE) was measured between the control sample (without the addition of steamed oyster mushroom) and the samples with its addition [75]. All measurements were made with SCI geometry, CIE L*a*b*C*h* color space, using an 8 mm diameter measurement aperture D65 illuminator and a CIE 10° standard observer. Before each measurement, the device was calibrated against a white standard.

#### 3.3.9. Statistical Analysis

All tests and experiments were conducted three times, and the results are presented as the average value ± the standard deviation (SD). Statistical analyses of the data were carried out using the Statistica 13.1 software (StatSoft Polska, Krakow, Poland). A significance level of *p* < 0.05 was used, and the analysis involved employing a one-way analysis of variance along with Tukey’s post hoc test [76].

The impact of the addition of oyster mushrooms on the antioxidant activity, oxidative changes, and selected sensory evaluation parameters in the final product was examined using Principal Components Analysis (PCA). The PCA was used for groups in which individual characteristics were most strongly correlated. The Statistica 13.1 program (StatSoft Polska, Krakow, Poland) was used for execution.

## 4. Conclusions

Replacement of some carp meat (5–20%) in fish burgers by mushroom (*Pleurotus osteatus*) is a solution that increases the nutritional value of the final product. Food fortification with bioactive substances is one of the trends in technology modification in order to promote the product and assign the properties of functional food. The addition of natural raw materials and non-isolated substances to achieve the intended goal is a much more difficult solution. The conducted experience proves that thanks to the development of such a technology, we obtain a product not only with the increased nutritional value of a functional food character but also significantly improving the sensory qualities of the finished product.

The antioxidant properties of oyster mushrooms were much higher than carp meat properties. In turn, the antioxidant properties of fish burgers were not as high as expected. Especially DPPH free radical scavenging ability and iron (III) ions reducing ability (FRAP) increased only a few percent when antioxidant activity against ABTS cation radical increased app. 20%. However, it must be noticed that lipid oxidation indices of burgers with oyster mushrooms were much lower than control burgers. Therefore, it can be assumed that one of the assumed goals, i.e., the protection of the lipid fraction through a statistically significant reduction in the total oxidation of lipids (TOTOX), has been achieved without the use of antioxidants. The antioxidant potential of raw materials positively influenced the quality of final product by limiting lipids changes during frying, and their antioxidant potential was still higher than the control sample.

Another advantage, apart from the nutritional value, is the fact that the addition of oyster mushrooms significantly improves the sensory attractiveness of the final product. As was shown in the article, a 15% addition of mushroom modifies the taste of the fish burger. Significantly improves the smell by reducing the intensity of the smell of fish meat. It improves the texture, improving the crispiness and delicacy of the product. The color of the fried fish burger received better notes in the sensory evaluation, which was confirmed by the instrumental color determination.

## Figures and Tables

**Figure 1 molecules-28-06975-f001:**
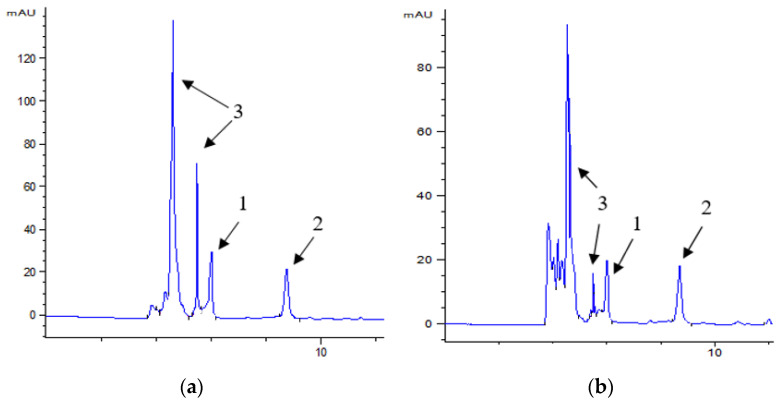
Chromatogram of raw (**a**) and steamed (**b**) oyster mushroom extracts; (1—gallic acid, 2—epicatechin; and 3—other peaks (unidentified).

**Figure 2 molecules-28-06975-f002:**
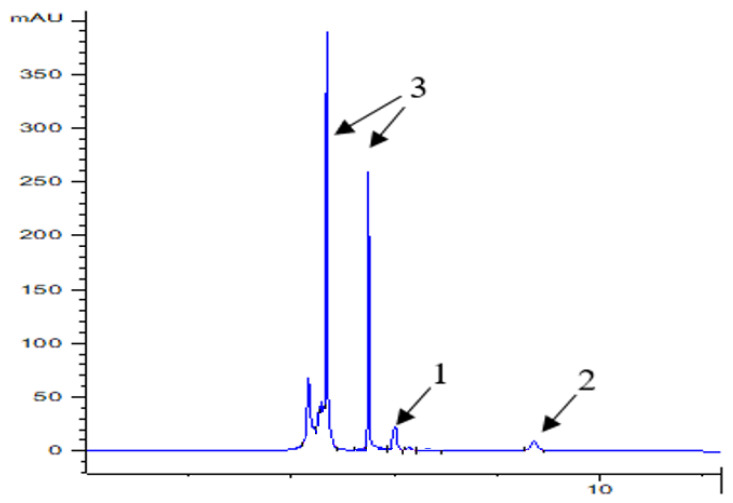
Chromatogram of fried burger extracts P20 (1—gallic acid, 2—epicatechin, and 3—other peaks (unidentified).

**Figure 3 molecules-28-06975-f003:**
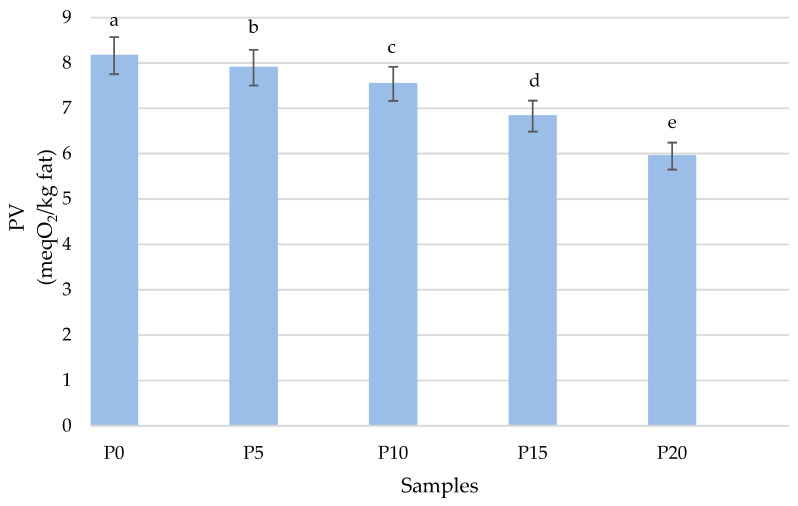
Comparison of changes in the peroxide value in fried burgers depending on the addition of oyster mushrooms. Means in a bar with the same lowercase letter do not differ significantly (*p* < 0.05).

**Figure 4 molecules-28-06975-f004:**
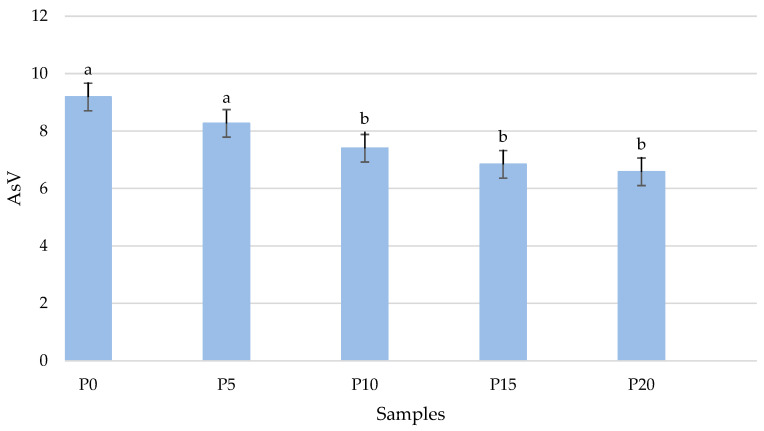
Comparison of changes in the anisidine value in fried burgers depending on the addition of oyster mushroom. Means in a bar with the same lowercase letter do not differ significantly (*p* < 0.05).

**Figure 5 molecules-28-06975-f005:**
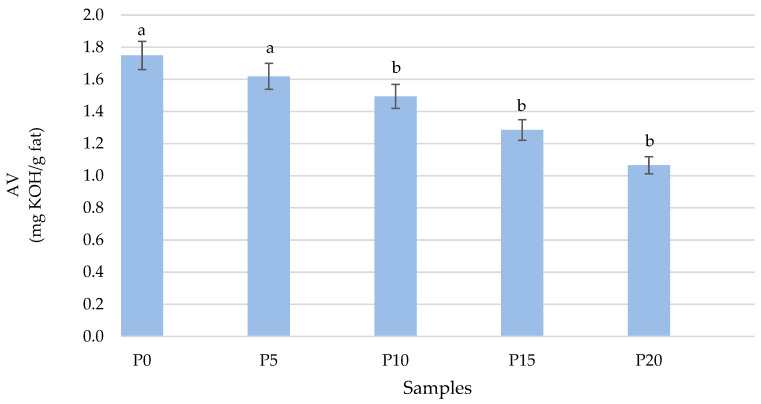
Comparison of hydrolytic changes in fried burgers depending on the addition of oyster mushrooms. Means in a bar with the same lowercase letter do not differ significantly (*p* < 0.05).

**Figure 6 molecules-28-06975-f006:**
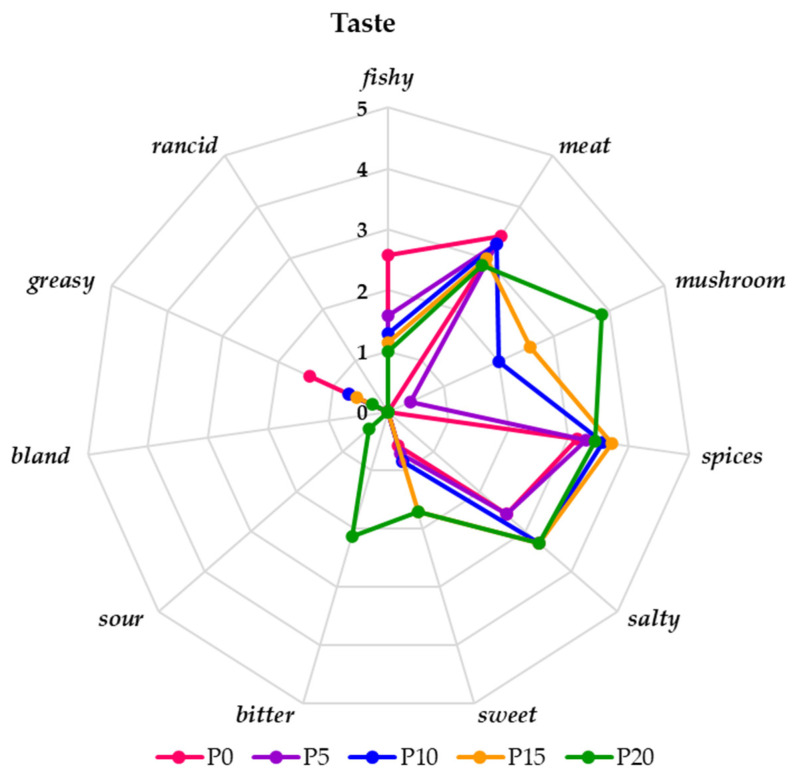
The taste profile of fried burgers with and without oyster mushrooms on a 6-point hedonic scale (0—imperceptible, 1—very weak, 2—weak, 3—moderate, 4—clear, and 5—very clear).

**Figure 7 molecules-28-06975-f007:**
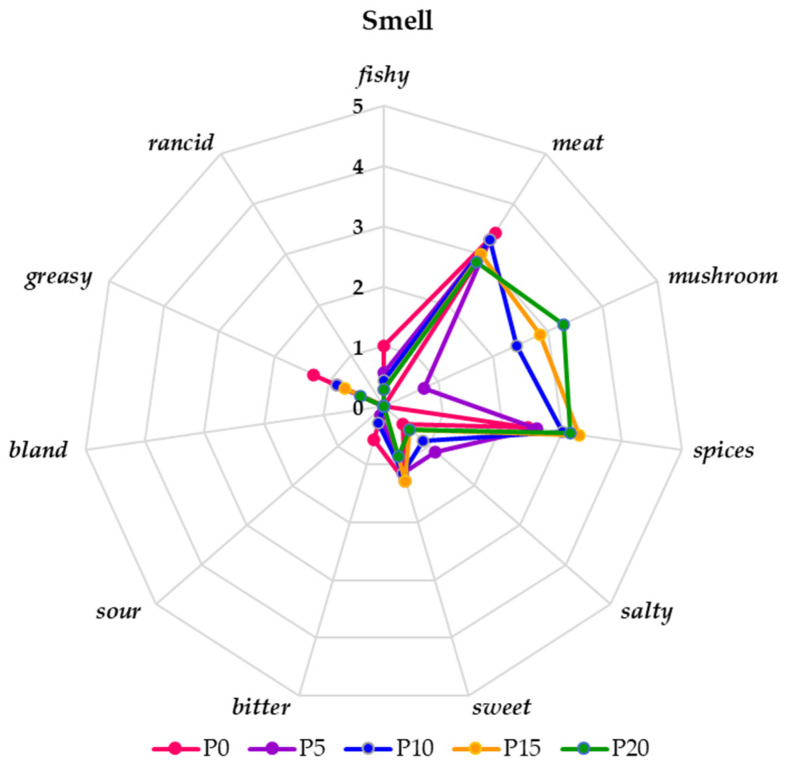
The smell profile of fried burgers with oyster mushrooms and without them on a 6-point hedonic scale (0—imperceptible, 1—very weak, 2—weak, 3—moderate, 4—clear, and 5—very clear).

**Figure 8 molecules-28-06975-f008:**
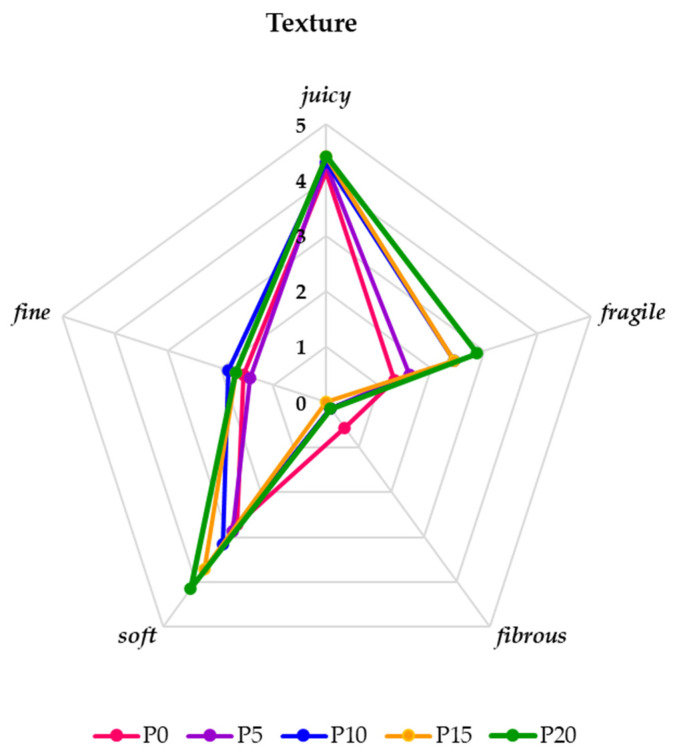
The texture profile of fried burgers with and without oyster mushrooms on a 6-point hedonic scale (0—imperceptible, 1—very weak, 2—weak, 3—moderate, 4—clear, and 5—very clear).

**Figure 9 molecules-28-06975-f009:**
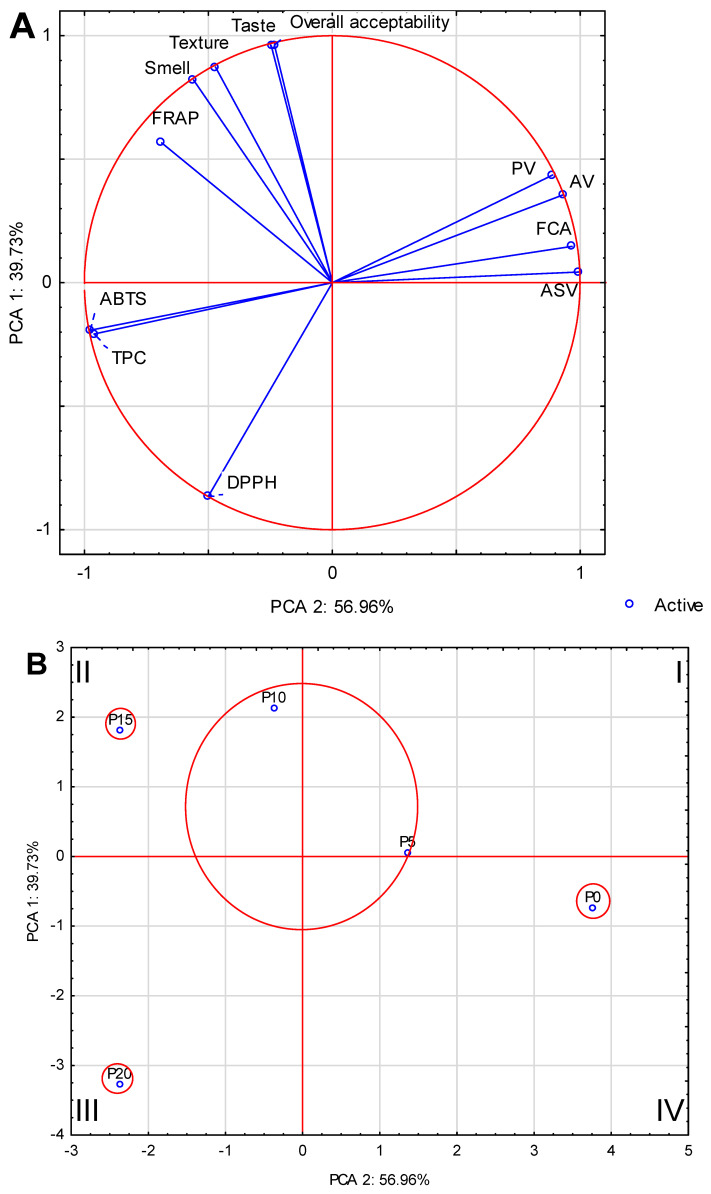
(**A**): PCA biplot of the first two principal components for antioxidant activity, detailed content, and sensory evaluation. (**B**): Component distribution of fried burgers based on two components from the principal component analysis.

**Table 1 molecules-28-06975-t001:** Basic composition of raw materials.

	Oyster Mushroom		Meat
	Fresh	Steamed	
Water [g/100 g]	91.26 ± 0.06 ^b^	89.45 ± 0.05 ^a^	73.98 ± 0.35
Proteins [g/100 g]	2.49 ± 0.083 ^a^	3.09 ± 0.104 ^b^	19.60 ± 0.60
Lipids [g/100 g]	1.35 ± 0.28 ^a^	1.15 ± 0.09 ^a^	4.89 ± 0.54
Ash [g/100 g]	1.159 ± 0.141 ^b^	0.693 ± 0.150 ^a^	0.907 ± 0.098

Means in rows with the same lowercase letter do not differ significantly (*p* < 0.05).

**Table 2 molecules-28-06975-t002:** Bioactive components, antioxidant properties, and oxidation indices of raw materials.

	Oyster Mushroom		Meat
	Fresh	Steamed	
Non-protein N [mg/100 g]	0.175 ± 0.038 ^a^	0.301 ± 0.006 ^b^	0.542 ± 0.002
α-amino N [mg/100 g]	54.6 ± 2.69 ^a^	98.3 ± 1.08 ^b^	95.7 ± 1.62
amino N [mg/100 g]	74.6 ± 1.4 ^a^	122.3 ± 1.5 ^b^	128.6 ± 1.8
PHP-Peptides [mg/100 g]	147.30 ± 5.30 ^b^	106.6 ± 7.3 ^a^	34.1 ± 0.7
PHP-Tyrosine [mg/100 g]	101.60 ± 0.80 ^a^	114.1 ± 3.2 ^b^	20.6 ± 0.2
TPC [mg GAE/g]	0.753 ± 0.001 ^b^	0.632 ± 0.001 ^a^	0.475 ± 0.005
ABTS [µmol TE/g]	4.68 ± 0.54 ^a^	4.11 ± 0.15 ^a^	1.97 ± 0.13
FRAP [µmol TE/g]	29.8 ± 1.1 ^a^	31.0 ± 2.1 ^a^	18.9 ± 1.2
DPPH [µmol TE/g]	0.607 ± 0.007 ^a^	0.604 ± 0.001 ^a^	0.290 ± 0.009
FCA [µmol EDTA/g]	0.537 ± 0.006 ^a^	0.618 ± 0.004 ^b^	2.227 ± 0.007

TPC (total phenolic compounds); GAE (gallic acid equivalent); ABTS (2,2′-Azino-bis(3-ethylbenzothiazoline-6-sulfonic acid) diammonium salt; TE (Trolox equivalent); FRAP (Ferric reducing antioxidant power); DPPH (2,2-Diphenyl-1-picrylhydrazyl); FCA (ferrous chelating ability); EDTA (ethylenediaminetetraacetic acid). Means in rows with the same lowercase letter do not differ significantly (*p* < 0.05).

**Table 3 molecules-28-06975-t003:** Comparison of epicatechin and gallic acid content in selected forms of oyster mushroom.

	Oyster Mushroom	
	Fresh	Steamed
Epicatechin [mg/100 g]	100.61 ± 3.18 ^b^	61.29 ± 1.7 ^a^
Gallic acid [mg/100 g]	34.24 ± 1.5 ^b^	5.53 ± 0.30 ^a^

Means in a row with the same lowercase letter do not differ significantly (*p* < 0.05).

**Table 4 molecules-28-06975-t004:** Oxidative and hydrolytic changes of basic components.

	Oyster Mushroom		Meat
	Fresh	Steamed	
PV (meqO_2_/kg fat)	4.116 ± 0.044 ^a^	5.020 ± 0.115 ^b^	6.433 ± 0.062
AsV	3.441 ± 0.100 ^a^	4.096 ± 0.108 ^b^	5.129 ± 0.151
AV (mg KOH/g fat)	0.308 ± 0.009 ^a^	0.392± 0.018 ^b^	0.574 ± 0.000

Means in a row with the same lowercase letter do not differ significantly (*p* < 0.05).

**Table 5 molecules-28-06975-t005:** Changes in fish burger mass after thermal treatment.

Sample	Weight before Frying [g]	Weight after Frying [g]	Losses [g]	Losses [%]
P0	74.45 ± 0.50 ^a^	67.52 ± 0.30 ^a^	6.93 ± 0.60 ^a^	9.31 ± 0.80 ^a^
P5	74.43 ± 0.60 ^a^	67.60 ± 1.10 ^a^	6.83 ± 0.30 ^a^	9.18 ± 0.40 ^a^
P10	75.18 ± 0.80 ^a^	68.35 ± 0.70 ^a^	6.83 ± 0.10 ^a^	9.08 ± 0.10 ^a^
P15	74.48 ± 0.60 ^a^	67.91 ± 1.40 ^a^	6.83 ± 0.90 ^a^	8.76 ± 1.20 ^a^
P20	75.25 ± 0.70 ^a^	68.57 ± 0.80 ^a^	6.68 ± 0.30 ^a^	8.88 ± 0.40 ^a^

Means in columns with the same lowercase letter do not differ significantly (*p* < 0.05).

**Table 6 molecules-28-06975-t006:** The basic composition of fried burgers with oyster mushrooms.

	P0	Burgers with Different Oyster Mushroom Addition			
		P5	P10	P15	P20
Water [g/100 g]	66.72 ± 0.26 ^a^	66.45 ± 0.22 ^a^	67.86 ± 0.03 ^b^	68.94 ± 0.17 ^c^	70.16 ± 0.17 ^d^
Proteins [g/100 g]	18.52 ± 0.09 ^c^	18.14 ± 0.23 ^c^	17.32 ± 0.074 ^b^	16.05 ± 0.05 ^a^	15.52 ± 0.39 ^a^
Lipids [g/100 g]	9.86 ± 0.22 ^b^	10.61 ± 0.14 ^c^	9.54 ± 0.19 ^b^	9.80 ± 0.16 ^b^	8.48 ± 0.09 ^a^
Ash [g/100 g]	2.092 ± 0.041 ^a^	2.094 ± 0.069 ^a^	2.296 ± 0.487 ^a^	2.015 ± 0.039 ^a^	2.072 ± 0.067 ^a^

Means in rows with the same lowercase letter do not differ significantly (*p* < 0.05).

**Table 7 molecules-28-06975-t007:** Bioactive components, antioxidant properties, and oxidation indices of fried burgers with oyster mushrooms.

	P0	Burgers with Different Oyster Mushroom Addition			
		P5	P10	P15	P20
Non-protein N [mg/100 g]	0.344 ± 0.020 ^a^	0.351 ±0.027 ^a^	0.348 ± 0.06 ^a^	0.353 ± 0.010 ^a^	0.295 ± 0.056 ^a^
α-amino N [mg/100 g]	126.9 ± 1.7 ^d^	195.7 ± 3.3 ^a^	207.8 ± 2.3 ^ab^	223.7 ± 1.4 ^bc^	236.6 ± 5.1 ^c^
amino N [mg/100 g]	405.3 ± 19.6 ^a^	402.0 ± 3.8 ^a^	396.2 ± 2.7 ^a^	395.9 ± 3.8 ^a^	385.8 ± 4.8 ^a^
PHP-Peptides [mg/100 g]	201.0 ± 5.8 ^b^	218.5 ± 0.0 ^b^	157.2 ± 3.2 ^b^	240.3 ± 3.4 ^d^	220.6 ± 23.8 ^b^
PHP-Tyrosine [mg/100 g]	62.9 ± 2.6 ^a^	66.9 ± 1.7 ^b^	70.3 ± 1.7 ^c^	64.6 ± 1.0 ^ab^	71.4 ± 2.0 ^c^
TPC [mg GAE/g]	0.610 ± 0.005 ^a^	0.621 ± 0.004 ^b^	0.627 ± 0.007 ^b^	0.650 ± 0.003 ^c^	0.654 ± 0.003 ^c^
ABTS [µmol TE/g]	2.24 ± 0.30 ^a^	2.54 ± 0.22 ^ab^	2.75 ± 0.18 ^bc^	2.93 ± 0.08 ^c^	3.14 ± 0.14 ^d^
FRAP [µmol TE/g]	33.9 ± 0.6 ^a^	35.4 ± 0.9 ^b^	35.8 ± 0.4 ^b^	35.7 ± 0.7 ^b^	35.0 ± 0.5 ^b^
DPPH [µmol TE/g]	0.475 ± 0.014 ^a^	0.479 ± 0.005 ^a^	0.472 ± 0.006 ^a^	0.477 ± 0.006 ^a^	0.507 ± 0.008 ^b^
FCA [µmol EDTA/g]	1.465 ± 0.001 ^c^	1.347 ± 0.002 ^b^	1.345 ± 0.003 ^b^	1.211 ± 0.002 ^a^	1.204 ± 0.004 ^a^

TPC (total phenolic compounds); GAE (gallic acid equivalent); ABTS (2,2′-Azino-bis(3-ethylbenzothiazoline-6-sulfonic acid) diammonium salt; TE (Trolox equivalent); FRAP (Ferric reducing antioxidant power); DPPH (2,2-Diphenyl-1-picrylhydrazyl); FCA (ferrous chelating ability); EDTA (ethylenediaminetetraacetic acid). Means in rows with the same lowercase letter do not differ significantly (*p* < 0.05).

**Table 8 molecules-28-06975-t008:** Comparison of epicatechin and gallic acid content in fried burgers with oyster mushroom.

	P0	Burgers with Different Oyster Mushroom Addition			
		P5	P10	P15	P20
Epicatechin [mg/100 g]	Not identified	3.05 ± 0.12 ^a^	6.11± 0.18 ^b^	9.15 ± 1.1 ^c^	12.89 ± 1.2 ^d^
Gallic acid [mg/100 g]	Not identified	0.24 ± 0.1 ^a^	0.51 ± 0.11 ^b^	0.77 ± 0.14 ^c^	1.44 ± 0.18 ^d^

Means in a row with the same lowercase letter do not differ significantly (*p* < 0.05).

**Table 9 molecules-28-06975-t009:** TOTOX values of fried burgers with oyster mushrooms.

	P0	Burgers with Different Oyster Mushroom Addition			
		P5	P10	P15	P20
TOTOX	25.35 ± 0.43 ^d^	24.08 ± 0.17 ^d^	22.37 ± 0.22 ^c^	20.50 ± 0.03 ^b^	18.53 ± 0.11 ^a^

Means in a row with the same lowercase letter do not differ significantly (*p* < 0.05).

**Table 10 molecules-28-06975-t010:** Mean values of sensory evaluation of fried burgers with the addition of steamed oyster mushrooms and their general acceptability on a 5-point scale (1—very undesirable, 2—undesirable, 3—neither undesirable nor desirable, 4—desirable, and 5—very desirable).

	P0	Burgers with Different Oyster Mushroom Addition			
		P5	P10	P15	P20
General appearance	4.00 ± 0.82 ^a^	4.57 ± 0.53 ^b^	4.86 ± 0.38 ^c^	5.00 ± 0.00 ^c^	4.14 ± 0.90 ^a^
Color	4.00 ± 0.82 ^a^	3.86 ± 0.69 ^a^	4.57 ± 0.53 ^b^	5.00 ± 0.00 ^c^	3.86 ± 0.90 ^a^
Taste	4.14 ± 0.38 ^b^	4.28 ± 0.65 ^b^	4.71 ± 0.49 ^c^	4.86 ± 0.38 ^c^	3.86 ± 0.69 ^a^
Smell	4.28 ± 0.49 ^a^	4.57 ± 0.53 ^a^	5.00 ± 0.00 ^b^	5.00 ± 0.00 ^b^	4.43 ± 0.53 ^a^
Texture	3.86 ± 0.69 ^a^	4.07 ± 0.61 ^a^	4.71 ± 0.42 ^b^	4.86 ± 0.38 ^b^	3.85 ± 0.69 ^a^
Overall acceptability	4.00 ± 0.58 ^b^	4.14 ± 0.38 ^b^	4.86 ± 0.38 ^c^	5.00 ± 0.00 ^c^	3.57 ± 0.53 ^a^

Means in rows with the same lowercase letter do not differ significantly (*p* < 0.05).

**Table 11 molecules-28-06975-t011:** Average values of objective color parameters of fried burgers with and without the addition of steamed oyster mushrooms.

	P0	Burgers with Different Oyster Mushroom Addition			
		P5	P10	P15	P20
L*	54.35 ± 0.76 ^a^	55.36 ± 0.39 ^b^	55.84 ± 0.07 ^b^	56.56 ± 0.32 ^c^	57.49 ± 0.21 ^d^
a*	10.38 ± 0.20 ^e^	10.06 ± 0.07 ^d^	9.61 ± 0.20 ^c^	8.63 ± 0.28 ^b^	8.12 ± 0.34 ^a^
b*	17.50 ± 0.87 ^e^	16.82 ± 0.16 ^d^	16.21 ± 0.23 ^c^	15.40 ± 0.50 ^b^	14.51 ± 0.50 ^a^
C*	20.74 ± 0.50 ^e^	19.27 ± 0.48 ^d^	18.59 ± 0.40 ^c^	17.84 ± 0.66 ^b^	16.97 ± 0.76 ^a^
H*	60.23 ± 0.29 ^e^	58.77 ± 0.17 ^d^	57.43 ± 0.19 ^c^	55.89 ± 0.32 ^b^	54.12 ± 0.20 ^a^
ΔE	………..	2.68 ± 0.48 ^a^	2.83 ± 0.10 ^a^	3.50 ± 0.20 ^b^	4.08 ± 0.09 ^c^

Means in rows with the same lowercase letter do not differ significantly (*p* < 0.05).

**Table 12 molecules-28-06975-t012:** Meat and oyster mushrooms share in burgers and sample codes (from P0—no participation to P20—20% share of mushrooms).

Ingredients	P0 85:0	P5 80:5	P10 75:10	P15 70:15	P20 65:20
Fish comminuted meat (%)	85	80	75	70	65
Oyster mushroom (%)	0	5	10	15	20

## Data Availability

Not applicable.

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
