# Peer review of "Effect of Oyster Mushroom Addition on Improving the Sensory Properties, Nutritional Value and Increasing the Antioxidant Potential of Carp Meat Burgers"

_molecules, 2023, doi:10.3390/molecules28196975_

Round 1

Reviewer 1 Report (Previous Reviewer 2)

L49-54 - you must clearly indicate whether this information applies to the Polish or global population

L87-92 - this fragment should not be included after the aim of the work. Moreover, there are no literature sources.

L120-125 - the sentence should be reworded in its current form is illogical

L209-214 - text to be redrafted, the lack of statistically significant differences means that they should not be significant from an economic point of view either

L314-317 - the sentence is illogical and should be reworded

L318, 319 – use acid value (AV) instead of free fatty acid

L438-…. - With regard to the L* parameter, use the term lightness instead of brightness

L442-443, 446-448, 453-455, 466-467, 469-470 - this information should be presented more precisely, e.g., The increasing share of oyster mushroom in the evaluated samples caused a decreasing in color saturation from 7% (in P5) to 18% (in P20) in relation to P0

L456- 457 - decreasing in the brightness of the L* parameter  - this phrase is illogical; it should be decreasing in the lightness (L*) or  decreasing in the L*

L524 - ± 1-2°C this entry is incorrect it should be ± 1°C or probably ± 2°C

L525 – (…) after cooling down ….- provide the temperature

L526 - Should sensory evaluation be carried out on chilled samples? Are you sure it's right?

L536-540- The authors should clearly indicate how many replications they performed/how many batches of each version of burgers were produced. The current text is ambiguous, it still indicates the number of repetitions of specific analysis

Author Response

Response to the Assistant Editor comments on manuscript ID: molecules- 2628396

Authors would like to thank for the comments to our journal submission entitled:  „Effect of oyster mushroom addition on improving the sensory properties, nutritional value and increasing the antioxidant potential of carp meat burgers”.  We appreciate the astute observations and constructive comments which helped us to improve our submission.

Changes made to the manuscript are marked in yellow, the response indicate exactly where the changes are.

Changes suggested by the editors are marked in green.

Reviewer I

Comment 1: L49-54 - you must clearly indicate whether this information applies to the Polish or global population

Response 1: The described information relates to the national scale, and more precisely, to aquaculture in Poland. Corrected as suggested by the reviewer (Chapter: 1.Introduction; L48-52).

C2: L87-92 - this fragment should not be included after the aim of the work. Moreover, there are no literature sources.

R2: The fragment was removed. Corrected as suggested by the reviewer (Chapter: 1.Introduction).

C3: L120-125 - the sentence should be reworded in its current form is illogical

R3: This sentence has been reworded. Corrected as suggested by the reviewer (Chapter: 2.1.2. Antioxidant properties and total polyphenols content; L120 – L126).

C4: L209-214 - text to be redrafted, the lack of statistically significant differences means that they should not be significant from an economic point of view either

R4: We suggested that any increase in productivity in industry may generate profits, hence our conclusion regarding the economic benefits despite the lack of statistically significant differences. Nevertheless, we agree with your comment and the sentence has been corrected. Corrected as suggested by the reviewer (Chapter: 2.2.1. Basic characteristic; L215-217).

C5: L314-317 - the sentence is illogical and should be reworded

R5: This is a valid point, the sentence was incorrectly formulated. The quality assessment of fish involves the determination of acid value, which can serve as an indicator not only for fish but also for other food products, such as fish burgers. In the case of hydrolytic changes, similar dependencies to those observed in oxidative changes can also be noticed. Corrected as suggested by the reviewer (Chapter: 2.2.3. Determination of lipid quality parameters; L322-324).

C6: L318, 319 – use acid value (AV) instead of free fatty acid

R6: The expression free fatty acid has been changed to acid value. Corrected as suggested by the reviewer (Chapter: 2.2.3. Determination of lipid quality parameters; L325-327).

C7: L438-…. - With regard to the L* parameter, use the term lightness instead of brightness

R7: Thank you for your attention. We agree with the opinion, it is a mistake. Changed brightness to lightness in whole article. Corrected as suggested by the reviewer (L446; L448; L463; L694).

C8: L442-443, 446-448, 453-455, 466-467, 469-470 - this information should be presented more precisely, e.g., The increasing share of oyster mushroom in the evaluated samples caused a decreasing in color saturation from 7% (in P5) to 18% (in P20) in relation to P0

R8: Thank you for your attention. Indeed, the sentences look better and are more precise. Corrected as suggested by the reviewer:

L447-448 The increasing share of oyster mushroom in the evaluated burgers resulted in an increase in color brightness by 2% (in P5) to approximately 6% (in P20) compared to P0.

L452-454 Generally, the increased addition of oyster mushroom in the evaluated burgers resulted in a reduction in the redness of the color by 3% (in P5) to 22% (in P20) compared to P0.

L460-461 In the evaluated burgers, with the increase in the addition of oyster mushroom, the yellowness of the color decreased by 4% (in P5) to 17% (in P25) in relation to P0.

L472-474 The increasing share of oyster mushroom in the evaluated samples caused a decreasing in color saturation from 7% (in P5) to 18% (in P20) in relation to P0.

L478-480 The increasing addition of oyster mushrooms in burgers resulted in a decrease in the H* parameter by 2% (in P5) to 10% (in P20) compared to the control sample.

C9: L456- 457 - decreasing in the brightness of the L* parameter  - this phrase is illogical; it should be decreasing in the lightness (L*) or  decreasing in the L*

R9: Thank you, that's a good point. We have corrected the meaning of the sentence. Corrected as suggested by the reviewer (Chapter: 2.2.5. Objective method of color parameters; L472-474).

C10: L524 - ± 1-2°C this entry is incorrect it should be ± 1°C or probably ± 2°C

R10: Thank you for your attention. The record has been corrected (170±1-2°C » 170±2°C and 75±1-2°C » 75±2°C). Corrected as suggested by the reviewer (Chapter: 3.2. Preparation of fish burgers; L533).

C11: L525 – (…) after cooling down ….- provide the temperature

R11: The temperature after the burgers cooled was 20±2°C, which corresponded to room temperature. The temperature is precisely specified in the text. Corrected as suggested by the reviewer (Chapter: 3.2. Preparation of fish burgers; L535).

C12: L526 - Should sensory evaluation be carried out on chilled samples? Are you sure it's right?

R12: Thank you for your attention. The text does not precisely elaborate on the conditions for carrying out sensory evaluation. The evaluation was carried out on warm burgers at a temperature of 62±2°C, and the analyzes were performed when the burgers were cooled down. This passage is explained in more detail in the text. Corrected as suggested by the reviewer (Chapter: 3.2. Preparation of fish burgers; L535-538).

C13: L536-540- The authors should clearly indicate how many replications they performed/how many batches of each version of burgers were produced. The current text is ambiguous, it still indicates the number of repetitions of specific analysis

R13: Thank you very much for attention. Probably not understanding the text may result from a bad translation into English. We hope that the text has been made more precise. Five versions of burgers were prepared in three separate batches, each yielding 12 patties. This approach allowed for three replications for each burger version. Additionally, the raw materials (carp meat and mushrooms) underwent testing five times. Corrected as suggested by the reviewer (Chapter: 3.3. Methods; L547-551).

Reviewer 2 Report (New Reviewer)

This manuscript brings very interesting topics, which are presented scientifically, but in easy-reading manner. 

Nevertheless, I ask the authors to correct the following:

1. The paper should be slightly shortened. Some results are extensively presented/explained, although they are given in the Tables. So it seems like doubling the results. For instance: l327-l357.

2. Way too much references (94) for the original research paper! Remove unnecessary references.

3. Wherever P < 0.05 is written, P should be in italic. Also p should be in italic in l164 (p-hydroxybenzoic acid), and l652 (p-anisidine). 

4. Correct the sentence (l614-7): Fish burgers and mushrooms WERE analyzed for gallic acid, chlorogenic acid, caffeic acid, ferulic acid, coumaric acid, catechin, epicatechin, rutin, kaempferol, apigenin, AND QUERCETIN, using the developed method. 

Actually, 'quercitin' should be replaced by 'quercetin' throughout the manuscript (l158, l616, l639).

5. Albeit the English is excellent, pay attention to some sentence constructions, for instance (l633): Total analysis time 60 minutes.

6. Once the abbreviation is introduced, it should not be repeated. Too many times the explanation of P0 is given - without the addition of oyster mushroom...

7. Chromatograms in Figure 1 and Figure 2 should be rearranged. Too crowded with peaks. So, shorten the x-axis (up to 12 minutes is enough) and/or expand the x-axis.  

8. What is the dry matter content (water content) of carp and steamed oyster mushrooms?

9. l228: Why new paragraph? 

Author Response

Response to the Assistant Editor comments on manuscript ID: molecules- 2628396

Authors would like to thank for the comments to our journal submission entitled:  „Effect of oyster mushroom addition on improving the sensory properties, nutritional value and increasing the antioxidant potential of carp meat burgers”.  We appreciate the astute observations and constructive comments which helped us to improve our submission.

Changes made to the manuscript are marked in yellow, the response indicate exactly where the changes are.

Changes suggested by the editors are marked in green.

Comment 1: The paper should be slightly shortened. Some results are extensively presented/explained, although they are given in the Tables. So it seems like doubling the results. For instance: l327-l357.

Response 1: Thank you for your attention. The expansion of the article was the result of suggestions from previous reviewers, hence the extensiveness of this article. The text has been shortened slightly, but we would like to meet the requirements of each reviewer to make the article valuable.

C2: Way too much references (94) for the original research paper! Remove unnecessary references.

R2: Thank you very much for attention. The literature was indeed very extensive. Literature reduced by 18 unnecessary references. Corrected as suggested by the reviewer.

C3: Wherever P < 0.05 is written, P should be in italic. Also p should be in italic in l164 (p-hydroxybenzoic acid), and l652 (p-anisidine).

R3: Thank you for your attention. The text was checked and fragments requiring correction were formatted (Whole article).

C4: Correct the sentence (l614-7): Fish burgers and mushrooms WERE analyzed for gallic acid, chlorogenic acid, caffeic acid, ferulic acid, coumaric acid, catechin, epicatechin, rutin, kaempferol, apigenin, AND QUERCETIN, using the developed method.

Actually, 'quercitin' should be replaced by 'quercetin' throughout the manuscript (l158, l616, l639).

R4: Thank you for your attention. The record has been corrected (was » were and apigenin, quercitin » apigenin, and quercetin) (L160; L616; L617; L641).

C5: Albeit the English is excellent, pay attention to some sentence constructions, for instance (l633): Total analysis time 60 minutes.

R5: Thank you for your attention. The entire text was re-checked and incorrect wording was corrected. Corrected as suggested by the reviewer (L363 and whole article).

C6: Once the abbreviation is introduced, it should not be repeated. Too many times the explanation of P0 is given - without the addition of oyster mushroom...

R6: This is a valid point. Changes have been made to the text and the explanation "without the addition of oyster mushroom" has been removed. Corrected as suggested by the reviewer (L328; L341; L378; L397)

C7: Chromatograms in Figure 1 and Figure 2 should be rearranged. Too crowded with peaks. So, shorten the x-axis (up to 12 minutes is enough) and/or expand the x-axis. 

R7: Figures 1 and 2 have been corrected and are now clearer (x axis shortened to approximately 12 min). Corrected as suggested by the reviewer.

C8: What is the dry matter content (water content) of carp and steamed oyster mushrooms?

R8: The water content in carp is less than 74%, while in steamed oyster mushroom it is approximately 89.5%. These values can be seen in table 1.

C9: l228: Why new paragraph?

R9: This is a valid point. There is no need for a new paragraph here - the fragments have been combined. Corrected as suggested by the reviewer (Chapter: 2.2.1. Basic characteristic; L230)

This manuscript is a resubmission of an earlier submission. The following is a list of the peer review reports and author responses from that submission.

Round 1

Reviewer 1 Report

Molecules

molecules-2508801

Effect of oyster mushroom addition on improving the sensory properties, nutritional value and increasing the antioxidant potential of carp meat burgers

Dear Editor,

The article deals with the evaluation of the effect of oyster mushroom addition on some quality parameters of carp burgers, with particular emphasis on the antioxidant effect. The manuscript has been generally well designed. However, Why was there a need to combine two valuable products (carp meat and mushroom)? The authors should explain this part very well. This oyster mushroom could be added to a less valuable product to increase its nutritional value. My specific comments and questions are below;

-       Give some information about carp meat burgers in terms of shelf life.

-       Line 422: Give more information about the frying conditions?

-       Line 476: w? we?

-       Lines 480 and 483: What kind of filter was used?

-       “Moisture” should be “water”. Because the results are higher than 50%.

-       Table 2: Check the results of the meat. The total content is higher than 100%!

-       Table 2: In terms of statistical evaluation, is a or c higher? Please check and uniform it!

-       Table 2 and 4: Why did the authors compare mushrooms with meat? In terms of mushrooms (fresh and steamed), it is ok. But remove the statistical evaluation for meat!

-       Lines 110 and 113: What could be possible reasons for that? Please discuss it.

-       Line 126: 70% of what?

-       Table 3: raw and fried burgers?

-       Table 4: raw and fried burgers?

-       Give method validation parameters for chromatographic analysis!

-       The authors are suggested to do correlation analysis or PCA for better explaining the results!

-       Discussion sections should be improved!

It looks good.

Author Response

Proszę zobaczyć załącznik.

Reviewer 2 Report

Manuscript title: Effect of oyster mushroom addition on improving the sensory properties, nutritional value and increasing the antioxidant potential of carp meat burgers

The subject of the work seems to be important, especially from a practical point of view and the authors should be congratulated on the idea. However, the manuscript has many disadvantages that should be carefully corrected, supplemented and explained

L48 - The statement that carp meat is a rich source of n-3 fatty acids is a bit of an exaggeration compared to other freshwater fish, it is at most a good source

L48-53 - does the information provided in this paragraph refer to the global or regional scale, e.g. Poland? these statements should be clarified

L72-75- This sentence needs to be rethought and reworded. Are fish burgers which may potentially contain many additional substances (can be highly processed) really preferred by people who value a healthy lifestyle? or  Is the reduction of fish fat the basic assumption of functional foods?

L76…….- Results and discussions section - Generally in the whole chapter, the results for the raw materials used should be presented and discussed first (oyster mushroom and carp meat), and then for the products made from them, i.e. burgers

L111 - humidity is probably not the best term in this context

L119 Table 3; L164 Table 5 etc…… - phrase in table headers ‘Burgers with fresh oyster mushroom addition’ is rather not correct it should be ‘Burgers with different oyster mushroom content/addition’ or ‘Burgers with steamed oyster mushroom addition’

L121-149 and L455-464 - in discussing the antioxidant properties of raw materials and products, rather use the radicals names against which these properties were determined. As you have noticed, basically all the results of antioxidant properties have been expressed Trolox equivalents (µTE) per 1 g of sample. In its current form, the entire discussion is very misleading

L121-149- This section lacks a reliable comparison of the obtained results of antioxidant properties and TPC with literature data. Were the obtained values for raw materials and possibly also for products similar, higher or lower?

L166-177 - based on the available literature, the main, most important polyphenolic compounds contained in oyster mushrooms should be indicated

L194 and L 203 - Figure 1. and 2 - peaks visible in the chromatogram, e.g. at 4.6 or 5.6 min, should probably be signed as unidentified or in some other alternative way

L201 - results for P0 burgers have ‘no data’ or simply ‘not identified’?

L244 - Is this sentence about the fatty acid profile or free fatty acids??

and L244-L25 -the authors probably do not fully understand the idea and the principle of determining the acid value (AV). They should use the phrase ‘free fatty acid’ everywhere. ‘(…) Similar results were observed by Nayak et al. (2015) [53], where a decrease in the percentage of fatty acids (…)’ a reduction in the percentage of fatty acids does not have to be related to a decrease in AV at all. AV shows level of free fatty acids released from triacylglycerols in the process of hydrolysis induced by various factors.

L283 and L340 - the term ‘consumers’ should be replaced with ‘panellists’. Sensory evaluation was carried out by a trained team, not consumers

L294-L351 – In this section, it should be indicating whether the observed differences were statistically significant or not

Table 11 – check carefully if letters next to means were placed correctly

L404 – 406 - give more details about mechanical separation of carp meat i.a. method of separation, period in which the raw material was obtained, how it was stored, how many individuals/fish, etc.

L411-423 - give more details about the preparation of burgers: hand or mechanically formed, what was the height of the disc, temperature/conditions in which the burgers were prepared, how there were cooked; deep-fried or pan-fried, in what type of oil, at what temperature, up to what internal temperature, etc. Were the analysis performed immediately after the burgers preparation or after a certain time?

How many replications of the experiment were performed

L427 - The ‘Extracts’ section currently looks at least strange. Some introductory sentence for what purpose the extracts were prepared for what analyses they were used for, etc. More details on the preparation of extracts should be introduced. Was the preparation of extracts only for burgers?

L435 - How this sentence should be understood? - requires clarification

L447 and L434 - why the lipid fraction was extracted by two different methods

L454 and L464 - give more details, what apparatus was used to perform the analysis, give the range of the standard curve etc.; results were expressed on a wet weight?

L501 - very complicated sensory analysis; many methods using different ranges of scales. In the future, it may be worth considering a more comprehensive sensory evaluation method, e.g. the QDA method

L525 - use the correct term – camera???

L530 - a one-way analysis of variance was probably performed before the post hoc test

L532 - The summary section should be significantly shortened and basically limited to the results obtained in the research

The linguistic correctness of the work should be carefully checked

The linguistic correctness of the work should be carefully checked

Round 2

Reviewer 1 Report

Dear Editor,

The authors have not very well replied my comments/questions. The most important issues in my previous evaluation are not present in the revised manuscript. Therefore, my comment is reject!

It needs minor revision.

Reviewer 2 Report

L* L* L*vv

The authors have significantly improved the manuscript, however, I have a few comments and suggestions that should be taken into account

L222-232 - Why has no effort been made to identify other phenolic compounds more specific to oyster mushroom since the literature data indicate their presence, which the authors mention in the L227-231 Why were the compounds selected for identification which are not mentioned in the literature for oyster mushroom?

L363-413 - the term ‘statistically’ is not needed, only significant/lly or insignificant is enough. Also provide the P-value or e.g. P<0.05 or P>0.05 if you mention significant or insignificant differences, respectively

L512-513- this sentence is redundant

L520-541 - in the revised manuscript and in the review response you state that the analyzes were performed in triplicate. Okay that seems reasonable. However, I asked you in how many replications did you perform the entire experiment, i.e. preparing all versions of burgers which is necessary to assess the variability of specific burger additives under different production conditions; in other words, to avoid the effect associated with only single batch

Table 11 - check once again the correctness of the letters next to means. They are still mislabeled in my opinion

The use of radical names alone is sufficient, additional terms such as TEAC RSA are completely redundant and very confusing

 Generally, your responses to review comments are not reviewer-friendly